



# Verification of the regional atmospheric model CCLM v5.0 with conventional data and Lidar measurements in Antarctica

Rolf Zentek and Günther Heinemann

Department of Environmental Meteorology, University of Trier, Germany

**Correspondence:** Rolf Zentek (zentek@uni-trier.de)

**Abstract.** The non-hydrostatic regional climate model CCLM was used for a long-term hindcast run (2002-2016) for the Weddell Sea region with resolutions of 15 and 5 km and two different turbulence parametrizations. CCLM was nested in ERA-Interim data. We prescribed sea-ice concentration from satellite data, and used a thermodynamic sea-ice model. The performance of the model was evaluated in terms of temperature and wind using data from Antarctic stations, AWS over land

and sea ice, operational forecast model and reanalyses data, and lidar wind profiles. For the reference run we found a warm bias for the near-surface temperature over the Antarctic plateau. This bias was removed in the second run by adjusting the turbulence parametrization, which results in a more realistic representation of the surface inversion over the plateau. Differences in other regions were small. A comparison with measurements over the sea ice of the Weddell Sea by three AWS buoys for one year showed small biases for temperature around 1 K and for wind speed of 1 $ms^{-1}$. Comparisons of radio soundings

showed a model bias around zero and a RMSE of 1-2 K for temperature and of 3-4 $ms^{-1}$ for wind speed. The comparison of CCLM simulations at resolutions down to 1 km with wind data from Doppler Lidar measurements during December 2015 and January 2016 yielded almost no bias in wind speed and RMSE of ca. 2 $ms^{-1}$. Overall CCLM shows a good representation of temperature and wind for the Weddell Sea region. These results encourage for further studies using CCLM data for the regional climate in the Antarctic at high resolutions and the study of atmosphere-ice-ocean interactions processes.

## 1 Introduction

Regional climate models (RCMs) are a valuable tool for improving our understanding of processes and interactions of the climate system in the polar regions. These processes are e.g. atmosphere-ice-ocean (AIO) interactions, which are particularly pronounced when sea ice formation is involved. This is associated with strong impacts on the surface energy fluxes and the atmospheric boundary layer (ABL). The added value of RCMs compared to coarser reanalysis and Global Climate Models

(GCMs) has been shown in a number of studies (e.g., Rummukainen, 2010) and is the background of the POLAR-CORDEX (COordinated Regional Downscaling Experiment) initiative (Akperov et al., 2018). For the polar regions, the spatial and temporal coverage by the observational network is sparse compared to mid-latitudes, therefore RCMs are the only means to provide climatological information at a high resolution with full spatial coverage (e.g., Kohnemann et al., 2017). High-resolution atmospheric simulations are also important for forcing ocean models (Haid et al., 2015), the understanding of the surface mass





balance (Souverijns et al., 2019; Gorodetskaya et al., 2014) and topographic effects such as foehn winds, which could play a role for the instability of ice shelfs (Cape et al., 2015) as well as katabatic winds (Ebner et al., 2014; Heinemann, 1997).

For the Antarctic, van Lipzig (2004) show that for a sufficient consideration of topography-induced atmospheric processes a resolution of at least 15 km is necessary. van Lipzig (2004) used the hydrostatic regional climate model RACMO (Regional

Atmospheric Climate Model) with 14 km resolution for the period 1987-1993. van Wessem et al. (2015) used also the RACMO model at a high resolution of 5.5 km over the period 1979–2013 for the Antarctic Peninsula (AP) and found more detailed and more pronounced temperature and wind speed gradients compared to the ERA-Interim forcing (approx. 80 km horizontal resolution), which are mostly related to the katabatic wind. However, the sea ice cover data set with 80 km resolution and the assumption that non-hydrostatic effects are small at 5 km resolution are drawbacks of that study. Foehn winds were studied by

Elvidge et al. (2015) particularly for the Larsen C ice shelf using the Met Office Unified Model at 1.5 km grid size. King et al. (2017) used model data from the Antarctic Mesoscale Prediction System (AMPS) with 5 km resolution for the summer season 2010/11 to study foehn wind effects also over Larsen C Ice Shelf. Turton et al. (2017) studied foehn effects over Larsen C Ice Shelf for May 2011 using the non-hydrostatic polar WRF model with 1.5 and 5 km resolution and find in general better results for the higher resolution. The latter studies were performed with non-hydrostatic models, but for rather short periods. The need

of non-hydrostatic models for high-resolution regional climate simulations is outlined by Giorgi and Gutowski (2015) and Prein et al. (2015).

In the present study the regional non-hydrostatic Consortium for Small-Scale Modeling (COSMO) model in Climate Mode [COSMO-CLM (CCLM)] is used to run simulations for the Antarctic with resolutions of ≈15 and ≈5 km for the time period 2002 to 2016. The simulation is forced with ERA-Interim reanalysis data and is the first long-term hindcast simulation with

20 a high-resolution non-hydrostatic regional climate model for the Weddell Sea region. The main purpose of the simulations is the study of AIO interactions in polynyas (see Ebner et al., 2014), which require a high resolution also in the sea ice data used as boundary conditions for the simulations. Thus we focus on the period since 2002, for which high resolution sea ice data from microwave satellite sensors are available (see Section 2). The CCLM data is also used as atmospheric forcing for a high-resolution sea-ice/ocean model (see Haid et al., 2015).

This dataset of atmospheric variables is compared to conventional measurements like radio soundings (RS) and both manned stations (MS) and automatic weather stations (AWS). Further an investigation is presented concerning the usage of Doppler wind lidar measurements in polar regions for verifications of model simulations. In section 2 the model and data sets used for the simulation and the verification are described, followed by the results of the verification (section 3) and finally the summary (section 4) and conclusions (section 5).

## 2 Data and Methods

### 2.1 CCLM

The CCLM is a regional non-hydrostatic model and is used as the community model for German climate research. It is a modified version of the COSMO model (version 5.0; Steppeler et al., 2003, http://www.cosmo-model.org; archived documen-





tation under zenodo (Zentek, 2019)) used by the Climate Limited-area Modelling (CLM) - Community (Rockel et al., 2008, http://www.clm-community.eu). Three different model setups are used for the simulations (see Table 1 and Fig. 1).

The first simulation with a resolution of ≈15 km (C15) is forced with ERA-Interim (Dee et al., 2011) for the time period 2002 to 2016 and the domain covers a quarter of Antarctica centered over the Weddell Sea. The second simulation with a

5 resolution of ≈5 km (C05) is nested inside the C15 domain and is only done for winter periods (Apr.-Sept.) 2002-2016. The third simulation (T15) uses the same setup as C15, but the turbulence parametrization was changed, since deficits in the C15 simulations were found for the stable boundary layer. In the T15 simulation, the minimal diffusion coefficients for heat and momentum were lowered (from 0.4 to 0.01 $m^2s^{-1}$) to allow for a very stable boundary layer over the Antarctic ice sheet during winter. Further, the parametrization of the impact of the inhomogeneity of the surface on the turbulent kinetic energy (TKE)

was modified. These modifications are based on the studies of Cerenzia et al. (2014), Hebbinghaus and Heinemann (2006) and Souverijns et al. (2019).

All simulations have a vertical resolution of 60 levels, that are terrain-following on the ground and gradually change into pressure following coordinates around 12 km height with the model top being at 25 km height. The runs were performed in a forecast mode, i.e. daily 24 hour simulations with 6 hours spinup to keep the hindcast close to reality. The model was adapted

to polar regions by the implementation of a thermodynamic sea ice model (Schröder et al., 2011), and a change of the albedo and snow parametrizations (Table 2). Further the RTopo2 data set (Schaffer and Timmermann, 2016; Schaffer et al., 2016) is used for the topography as the default data set of CCLM did not include ice shelves. Parameters for the subgrid-scale orography (SSO, Lott and Miller, 1997) module were computed for the new dataset and the SSO module was used for both the 15 and 5 km simulation.

For sea-ice data, daily sea ice concentration (SIC) is used. The data is based on AMSRE and AMSR2 (Advanced Microwave Scanning Radiometer for Earth Observing System / 2) and for data gaps SSMIS (Special Sensor Microwave Imager / Sounder) satellite measurements (Spreen et al., 2008; Ezraty et al., 2007) are used. The resolution of the sea ice concentration data is 6.25 km for AMSRE/AMSR2, but coarser for SSMIS (12.5 km). Details for the used data are given in Table 3. Sea surface temperature (SST) data and initial surface temperature were taken from ERA-Interim. In the case of inconsistency between

SST and SIC (surface temperature below the freezing temperature (-1.7°C), for a SIC of 0%), the SST was set to the freezing temperature. The SIC data included some missing values that were replaced in the following way. In a first step missing values were filled with values from the day before and after (mean if both were available). In a second step, days where no data was available were interpolated linearly in time (overall 35 days; maximal 9 in succession). This still left some missing values (mostly along the coastline due to the different land masks of RTopo2 and AMSRE/SSMIS/AMSR2). These remaining missing

values are filled in a third step with an iterative procedure for each day separately using the surrounding grid points.

As daily sea ice thickness data like PIOMAS (Zhang and Rothrock, 2003) is not available for Antarctica, a sea ice thickness of 0 m / 0.1 m / 1 m is assumed for grid points where the sea ice concentration is 0-15% / 15-70% / 70-100%. This estimate is reasonable for the Weddell Sea (Kurtz and Markus, 2012). With a threshold of 70% SIC commonly used for the identification of polynyas, this choice is in accordance with previous studies (Ebner et al., 2014; Bauer et al., 2013).



## 2.2 AMPS and ERA

Beside the forcing data set ERA-Interim (Dee et al., 2011), the newer ERA5 reanalysis data (Hersbach et al., 2018) and data from the Antarctic Mesoscale Prediction System (AMPS, Bromwich et al., 2005; Powers et al., 2012) is used for comparisons. ERA5 reanalysis data is the new version of ERA-Interim reanalysis. Both data sets are products of the European Centre for Medium-Range Weather Forecasts. The AMPS data set was produced as a collaborative effort between Mesoscale and Microscale Meteorology Laboratory of the National Center for Atmospheric Research and The Ohio State University. The horizontal/temporal resolutions are approximately 80 km/6h (ERA-Interim), 30 km/1h (ERA5) and 10 km/3h (AMPS).

## 2.3 Surface measurements

We use near surface temperature and wind measurements from manned stations (MS) and automatic weather stations (AWS). The location of used MS and AWS are shown in Fig. 1 (circle / triangle) and detailed information are given in Table 4. The data was collected by the National Antarctic Operators and collated by the British Antarctic Survey (ftp://ftp.bas.ac.uk/src/SCAR_EGOMA).

Because maintenance of AWS is difficult for logistic reasons, they are more likely to include measurement errors. Thus we used the data from MS whenever possible and only fell back to AWS data for regions where no MS was available. An examination of the data showed some obviously wrong data where the wind speed drops e.g. from from 15 $\mathrm{ms}^{-1}$ to 0 $\mathrm{ms}^{-1}$ between two data records. As there were also longer periods even over days during which the data showed 0 $\mathrm{ms}^{-1}$, we refrained from searching for these drop-offs with a threshold and instead removed all wind data with a wind speed of 0 $\mathrm{ms}^{-1}$. This removed less than 8% of the data for each station except for three manned stations (Belgrano II, Esperanza and San Martin) where up to 35% were removed. Furthermore the wind direction for the years 2002-2005 of the Larsen AWS were removed as there seemed to be an offset compared to all following years.

As this MS and AWS data set lacks observations over the ocean and sea ice, we also used another dataset of three AWS (Grosfeld et al., 2016) that where placed on ice floes and cover each a timespan of about one year. As they were placed on ice floes, these AWS drifted through the Weddell Sea from January to December 2016. The locations are shown in Fig. 2. For this data set we only removed 4 outliers where longitude and latitude was obviously wrong. Further the last 31 data points from AWS 3 were removed as the AWS 3 data stops in December and a corruption in the end is very likely.

## 2.4 Radio soundings

To assess the model performance in the whole atmosphere, radio sounding (RS) data was downloaded from the University of Wyoming (http://weather.uwyo.edu/upperair/sounding.html). The location of RS are shown in Fig. 1 (filled squares) and detailed information are given in Table 5. Some RS had an unrealistic pressure value at a given height. To remove these, we checked if the deviation from the mean pressure was bigger than three times the standard deviation for that height. This removed only 2-3% of the RS. Further we only selected RS done at either 00:00 UTC for Amundsen-Scott and Novolazarevskaya or at 12:00 UTC for Halley, Marambio, Neumayer and Rothera, because these were the only times when the RS were done regularly.





## 2.5 Lidar

In the austral summer 2015/2016 we conducted in-situ measurements in the Weddell Sea region. We installed a Doppler lidar onboard the RV *Polarstern* and measured vertical profiles of horizontal wind speed and direction from 24th December till 30th January. In Zentek et al. (2018) we compared the measurements to radio soundings and ship measurements and found a bias

(root mean square deviation) of approx. 0.1 (1) $\mathrm{ms}^{-1}$ for wind speed and 1 (10)° for wind direction, respectively. Lidar wind profiles are available with a vertical resolution of 10 m and with a temporal resolution of ca. 15 minutes. For the comparison, profiles were average to hourly values and 50 m height resolution (Zentek and Heinemann, 2019a).

For the purpose of comparisons we also set up another model domain with a 1 km resolution and nested it inside the 5 km domain. We ran both with the original settings (C01/C05) and changed turbulence parameters (T01/T05) for the measuring

period (see Table 1).

## 2.6 Methods

For the comparison of CCLM with AMPS and ERA-Interim data, the latter were interpolated bilinearly to the CCLM grid points. For the comparisons to measurements (MS, AWS, RS and lidar) the nearest neighboring grid point of CCLM was selected. For surface stations, the CCLM temperature was corrected with 1 K per 100 m for the height difference of the station

and the respective grid point (see Table 4 and 5 for information on grid point heights and difference to the actual station height).

For the radio sounding comparisons, we made a vertical linear interpolation of model and radio sounding data to the same pressure level (equidistant every 50 hPa). Only data at a certain pressure level was analyzed if the amount of measurements was more than half the median of the number of observations over all heights. Prior to the calculation of the correlation for temperature, monthly means were subtracted to remove influence from the seasonal cycle.

In case of the three AWS buoy on ice floes, the wind speed was measured at a height of approximately 2 m. We therefore assumed a logarithmic wind profile and neutral stability with a roughness length of 0.001 m and thus scaled CCLM 10 m wind speed by a factor of 0.825 to calculate the 2 m wind speed. For the AWS over land no correction was applied as the height of sensors was uncertain or unknown.

For the lidar comparisons we interpolated model, reanalyses and lidar data to an equidistant grid (height every 50 m up to

25 1000 m). As ERA-Interim only has output every 6 hours we did not interpolate linearly in between, to have a sharper distinction to ERA5. Further note that the lidar data is an average over one hour around every full hour, which smooths the data and makes it better comparable to the simulation data that represent the wind average over the whole model grid box.

The wind comparisons are based on the magnitude of wind speed and the wind direction (no vector differences) unless stated otherwise. For wind direction we always assume a maximal possible difference of 180° and removed cases where wind speed

is lower than 0.5 $\mathrm{ms}^{-1}$. We compute the root mean square error (RMSE) and use the Pearson correlation coefficient (Corr) for temperature and wind speed, but use an adapted version for angular variables (Jammalamadaka and Sarma, 1988) (circ.Corr) for wind direction.





## 3 Verification

### 3.1 Model and reanalyses

In the first analysis the near-surface variables of CCLM are compared with ERA-Interim, ERA5 and AMPS. We computed monthly mean values over the period of 2002-2016 of 2 m temperature and 10 m wind speed. As the data sets of AMPS
(with the latest configuration) does not cover the whole period, we selected the years 2014-2016 for the main comparisons. For ERA-Interim we show both time periods.

The 2 m temperature bias for C15 for the winter (Apr.-Sep.) and summer (Jan.-Mar. and Oct.-Dec.) is shown in Fig. 3. The bias for summer is small and acceptable. For winter C15 is 1-3 K colder over sea ice than ERA5 and ERA-Interim, but this is still an acceptable difference. Over the East Antarctic Plateau (topography approximately higher than 2 km), a large
temperature bias up to 8 K compared to ERA5/ERA-Interim and up to 15 K compared to AMPS is visible during winter. The too warm temperature of CCLM over the plateau could be attributed to a too strong mixing in the surface boundary layer. This was the reason for changing the turbulence parametrization (T15).

As the change of turbulence parameters allows for more stable atmospheric boundary layer, T15 is overall colder than C15 near the surface, but this influence is very weak during summer or over the sea ice. The 2 m temperature bias for T15 is shown
in Fig. 4. Over land and especially over the East Antarctic Plateau the strong bias in winter present in C15 is reduced in T15 compared to AMPS and even turns into a negative bias compared to ERA5 and ERA-Interim. Fig. 5 shows the 10 m wind speed bias for C15 for the summer and winter period. The bias for T15 is very similar (see supplement; Fig. S1). Compared to ERA5 and ERA-Interim, C15 shows stronger winds (up to 5 ms$^{-1}$ more) over the Antarctic Peninsula and in the katabatic wind areas. For the winter period C15 simulates slightly weaker winds over the northern part of the sea ice when compared to ERA5 and
ERA-Interim, which may be a result of the different sea-ice parametrization. The bias of C15 compared to AMPS is mainly negative over the ice sheet, and a slight positive bias can be seen only for the Filchner-Ronne Ice Shelf. The C05 simulation (not shown) shows slightly higher 10 m winds (1 ms$^{-1}$) compared to the C15 and slightly lower (1 K) 2 m temperature.

Overall the C15 simulation is comparable to reanalyses and AMPS model data except for the large positive temperature bias during winter over high topography. When using the modified turbulence scheme, the bias with respect to the reanalyses is
25 reversed, but the agreement with AMPS is improved.

### 3.2 AWS and surface stations

To further investigate the differences between CCLM and other simulations from the last section, we compared C15 and T15 with surface measurements. The selection of stations was done after a quality check and using only stations with sufficient record length. In addition the stations should represent typical areas of the Weddell Sea region. The locations of the selected
stations are shown in Fig. 1 and detailed information is given in Table 4.

An 10 day comparison of measurements and CCLM model output at the station Halley is shown in Fig. 6. Both C15 and T15 capture the daily cycle of temperature, but T15 underestimates the temperature during some nights with low wind speeds.





Wind speed and direction of C15 and T15 are similar and agree very well with the measurements. Only during the first day the change of wind direction is different but the wind speed for this day is also very low.

For the full comparison of C05, C15 and T15 with all stations we calculated monthly bias, RMSE and correlation for winter and summer separately. Statistics for 2 m temperature are shown in Fig. 7.

The problem of the temperature bias of C15 over the plateau can be demonstrated for the Amundsen Scott data (No. 1). The +8 K bias for C15 in winter is reduced to less than 1 K in case of T15, thus showing the better performance of T15. The improvement can be seen also for summer. On the other hand a small cold bias is present for T15 for the coastal region. The statistics for 10 m wind speed (Fig. 8) and direction (Fig. 9) show almost no difference between C15 and T15. The reduced bias of T15 compared to C15 in wind direction for Amundsen Scott (No. 1) is a result of the better representation of the stable

boundary layer in T15. This yields colder surface temperatures that allows for a stronger wind shear and thus a reduced wind direction bias.

At AWS Union (No. 2) wind direction is almost constant with time which results in a low correlation although the bias and RMSE are comparable to other stations. For AWS Fossil (No. 8) there are two dominant wind directions both meassured and simulated, but they do not allways coincide in time and thus the RMSE is also very high.

The strong bias in wind direction for Bellinghausen (No. 16) is likely explained by the different small-scale topography around the stations, which not captured at the model resolutions. Also a data error of the station cannot be ruled out, as the other northern Antarctic Peninsula stations are relatively close to each other and do not show this bias. The reasons for the high bias and RMSE of wind direction for Belgrano II (No. 3) are also likely a result of small-scale topography effets.

Overall CCLM has a tendency to perform slightly better during summer and differences between the model runs C05, C15

and T15 are only visible in case of 2 m temperature. When calculating daily instead of monthly bias, RMSE and correlation, the results are similar, but show a much higher variance. These statistics are shown in the supplement (Fig. S2, S3 and S4).

In the last section (3.1), biases in temperature and wind speed were found compared to AMPS, ERA5 and ERA-Interim over sea ice. Observations over sea ice are rare, but the three drifting AWS buoys allow for a comparison for a full yearly cycle for the year 2016. All buoys were deployed in January 2016 near the east coast of the Weddell Sea, but at different positions. The

buoys No. 1 and 3 drifted from their original position near the coast of northwards out of the Weddell Sea and No. 2 stayed near the east coast (see Fig 2). An overview of the measurements for the AWS3 buoy is shown in Fig. 10. The seasonal cycle in temperature is captured by all model runs and wind speed and direction agree well.

The bias and RMSE of CCLM based on hourly temperature and wind speed for all AWS is given in Table 6 and 7. Overall AWS1 and AWS3 show similar statistics as both drifted relative synchronously northwards while AWS2 stayed close to the

coast north of the station Halley (No. 4). C15 shows a temperature bias of -0.3/-0.8 K for AWS1/AWS3 during winter, while T15 shows a slightly larger bias of -1.4/-1.7 K. This is not as high as the previously seen cold bias over sea ice during winter of CCLM compared to ERA-Interim and ERA5 of -2 K for C15 and -3 K for T15 (see Fig. 3 and Fig. 4). The RMSE is approx. 4 (3) K during summer (winter). For wind speed the RMSE is around 1.5 to 2 $\mathrm{ms}^{-1}$ and biases are equal or smaller than 0.7 $\mathrm{ms}^{-1}$ during summer and a little higher around 1 $\mathrm{ms}^{-1}$ during winter (see supplement; Fig. S5 and S6).



### 3.3 Radio soundings

The location of the radio soundings are shown in Fig. 1 as black squares. Note that Novolazarevskaya is very close to the model boundary (8 grid points) and CCLM may be partly influenced by the ERA-Interim boundary data. The radio soundings are done regularly at 00:00 UTC (= 6 hours after model start) for Novolazarevskaya and Amundsen Scott and 12:00 UTC (= 18

5    hours after model start) for Marambio, Neumayer, Rothera and Halley. The profiles of the temperature statistics (Fig. 11) show almost no bias except below 800 hPa and the RMSE is around 1 K in the upper troposphere for the coastal stations. The bias is slightly lower for C05 (only winter) and for C15 in summer. The correlations are larger than 0.8. These results are similar to the findings of Souverijns et al. (2018) that show a mean average error of 0.5 to 1.4 K. For Amundsen Scott (f) a large positive bias and a large RMSE is present in the lowest layers, which is most pronounced in winter. While for the winter the RMSE and the

correlation above 500 hPa is comparable to the coastal stations, a larger RMSE and correlations of less than 0.75 are present above 500 hPa during summer. The higher resolution of C05 yields only slight improvements for Marambio (a) and Rothera (d) at the Antarctic Peninsula, where the influence of the topography is larger than at the other stations. We did not include T15 in Fig. 11 as the statistics were almost identical to C15 with the exception of the lowest levels for Amundsen Scott. To address the differences of C15 and T15, a comparison of the mean temperature for the lowest 1 km of the atmosphere is shown

in Fig. 12. The changed turbulence parametrization only influences the cases of strong surface inversions. For Amundsen Scott (f) there is a clear improvement of T15 for the mean SBL structure during winter, but also during summer. Similar but weaker improvements can be seen for the eastern Weddell Sea (Halley (e) and Neumayer (b)). However, for Novolazarevskaya (c) and Rothera (d) a stronger bias in the lowest 100 m is present for T15. Above the surface inversion, differences for C05, C15 and T15 and the summer and winter season are relatively small, with only a minor exception of a small increase of RMSE above

500 hPa for Amundsen Scott (f) during summer.

For the comparison of wind speed (Fig. 13) and direction (Fig.14) we excluded T15 again, as it was almost identical to C15. The bias is again almost zero except near the surface. The RMSE for wind speed is around 3 to 4 ms$^{-1}$ and slightly lower during summer. Bias and RMSE are largest for Marambio (a) and Rothera (d) in the lowest 200 hPa, and as for the temperature C05 yields slight improvements for these stations. Souverijns et al. (2018) found a mean average error for wind speed of 2.1 to

3.6 ms$^{-1}$ for all seasons. The RMSE for wind direction is around 50° near the surface and reduces with height to 20° at 250 hPa, except for Amundsen Scott (f) where it stays around 50°.

### 3.4 Verification using wind lidar data

Wind profile measurements from lidar data are available for 24 December 2015 to 30 January 2016. We selected two case studies for comparisons. The first one features the occurrence of three low level jets (LLJs) during one "night" and the fol-

lowing morning. The second case study gives an overview of the differences and similarities between lidar measurements and simulations during a 10 day period.





### 3.4.1 Overall statistics

We also computed the overall statistics for all available lidar measurements (see Table 8). The different CCLM runs are very similar with no or only very small bias in wind speed and a RMSE of around $2\,\mathrm{ms^{-1}}$. For wind direction there was a small bias of -5° present and an RMSE of 30°. ERA5 and ERA-Interim show similar values. This good agreement could stem partially

from the fact, that the radio soundings of the ship (2-3 per day) are assimilated in ERA, which show also a good agreement with the lidar data (Zentek et al., 2018). The computation of the statistics for different heights showed that the wind speed RMSE of CCLM is largest around $1000\,\mathrm{m}$ height, while the RMSE of ERA5 and ERA-Interim is mostly constant with height (see supplement; Fig. S7).

### 3.4.2 Case study A

10 During the night from 16 Jan. 2016 to 17 Jan. 2016 the RV *Polarstern* operated in a polynya in the lee of the iceberg A23 (see Fig. 1). Three LLJ events were observed with the lidar (Fig. 15). The first LLJ occurred between 00:00 UTC and 02:00 UTC (LLJ1). The LLJ between 06:00 and 08:00 UTC (LLJ2) was captured by the radio sounding at 07:00 UTC (Zentek et al., 2018), and the wind maximum between 10:00 and 14:00 UTC (LLJ3) was also measured by a radio sounding in 800 to 1000 m height at 12:00 UTC. While the 6 hourly ERA-Interim data cannot reproduce the structure and evolution of the wind field of the lidar

measurements, the hourly ERA5 data capture LLJ2 and LLJ3, which is likely explained by the assimilation of radio sounding data. However, the LLJ wind speeds are underestimated and LLJ1 is missing in ERA5. The CCLM simulations (nested in ERA-Interim) show that the increase of resolution yields increased wind speeds particularly for LLJ3, but the height of the LLJ is too low. An indication of LLJ1 is seen in the CCLM simulations, but the wind speed is underestimated. The overall pattern of the wind direction field is well reproduced by all CCLM simulations. Since the ship's position was not stationary

for this period, we also tested for the dependency on the chosen grid point of the model, by choosing one grid point over the iceberg A23 and in the middle of the open polynya instead of the ship location. This had only a small effect and we therefore concluded that all the changes and patterns are mostly time and height dependent.

### 3.4.3 Case study B

From 20 to 30 Jan. 2016 RV *Polarstern* was navigating around the area of the Brunt Ice Shelf (see Fig. 1). The days show a

25 broad variety of different wind patterns (Fig. 16) ranging from no wind (on the 21th) to wind speeds exceeding $20\,\mathrm{ms^{-1}}$ (on the 29th) and also featuring vertically inhomogeneous winds both in speed and direction (on the 24th-26th). On the scale of days T15, ERA5, ERA-Interim and the lidar show the same evolution of the wind field. On smaller scales CCLM and the lidar show more detail, but CCLM does not always agree well with the lidar (e.g. on the 26th). ERA5 agrees well with the lidar data and sometimes even catches the small-scale details of measured wind patterns (e.g. on the 27th). T05 and T01 are very similar

to T15, with only little increased wind speeds (see supplement; Fig. S8).





If we presume that the lidar measurements are representative for the winds in the whole area that is covered by the model grid box, this case study gives a good impression on how reliable reanalyses and models are on those scales: e.g. for a simulated LLJ we cannot always assume that a LLJ was really present, even if the overall RMSE is shown to be smaller than 3 ms$^{-1}$.

## 4   Summary

We used the non-hydrostatic model COSMO-CLM (CCLM) nested in ERA-Interim data to produce a long-term hind cast (2002-2016) for the Weddell Sea region with resolutions of approximately 15 and 5 km and two different turbulence parametrizations. Sea-ice concentration is prescribed from satellite data and a thermodynamic sea-ice model is used. In this paper we evaluated the performance of the model in terms of temperature, wind speed and direction using data from Antarctic stations and AWS over land and sea ice. Comparisons to AMPS model and reanalyses data showed good agreement, except for a large

difference in surface temperature over the Antarctic plateau. The warm bias is also found in comparison to measurements at the Amundsen Scott station (surface and radio sounding), where the reference run C15 showed a strong warm bias near the surface (+8 K). This bias to station and AMPS data was removed in the second run T15 by adjusting the turbulence parametrization, which results in a more realistic representation of the surface inversion over the Antarctic plateau. But this caused also a small cold bias (down to -4 K) for other surface stations located on ice shelves in the eastern Weddell Sea. Differences in other

regions were small. A comparison with measurements over the sea ice of the Weddell Sea done by three AWS buoys for one year showed small biases for temperature around ±1 K and for wind speed of 1 ms$^{-1}$.

Comparisons with radio soundings showed a model bias around zero for all model levels except near the surface. In general, a RMSE of 1-2 K for temperature and of 3-4 ms$^{-1}$ for wind speed was found.

The comparison of CCLM simulations at resolutions down to 1 km with wind data from Doppler lidar measurements during

December 2015 and January 2016 in the southern and eastern Weddell Sea yielded almost no bias in wind speed and an RMSE of ca. 2 ms$^{-1}$. For wind direction the bias was ca. -5° with a RMSE of around 30°. Overall, CCLM is able to produce realistic evolution and structures of the wind in the ABL, but for specific events like LLJs differences occur in the location.

## 5   Conclusions and outlook

CCLM shows a good representation of temperature and wind for the Weddell Sea region. The adjustment of the turbulence

parametrization for very stable conditions is important for the realistic representation of the surface inversion over the Antarctic plateau. Since verification data for simulations are rare in the Antarctic, new types of measurements like Doppler lidar or controlled meteorological balloons (Hole et al., 2016) can give additional insights in the performance of atmospheric models. For the comparisons of CCLM with ship-based Doppler lidar in the present study the benefit of CCLM compared to ERA5 is small due to the fact that the ship's data were assimilated in the reanalysis and effects of topography were small. A larger benefit

is seen for polynya areas and the Antarctic Peninsula with small-scale topography. The YOPP (Year of Polar Prediction) project





will lead to more and enhanced observational data, which can be used for further verifications in the future. Future work with CCLM will be the study of atmosphere-ice-ocean interactions processes and quantification of sea ice production in polynyas.

*Code and data availability.*  The COSMO-CLM model is completely free of charge for all research applications. The current version of the COSMO-CLM model is available from the CCLM website: https://www.clm-community.eu under the licence http://www.cosmo-model.org/
content/consortium/licencing.htm. The particular version of the CCLM model used in this study is based on the official version 5.0 with additions to the sea ice module (according to Schröder et al. (2011)) and the changes in the turbulence parameterizations described in this study. If eligible access can be granted to the model source code under zenodo (Zentek and Heinemann, 2019b). The model output used in this study is archived under zenodo (Zentek and Heinemann, 2019c). The full model output data will be archived for a limited amount of time and are available on request (zentek@uni-trier.de). The model documentation is archived under zenodo (Zentek, 2019). The scripts
and configurations to run the simulations are archived under zenodo as well (Zentek and Heinemann, 2019d). The scripts to analyse the simulations and produce the figures in this paper are archived under zenodo as well (Zentek and Heinemann, 2019e).

*Author contributions.*  RZ carried out the setup of the model, simulation, data curation, methodology, validation, visualization and writing of the original draft. GH carried out the conceptualization, methodology, a review of the writing, editing, supervision, project administration and funding acquisition.

*Competing interests.*  The authors declare that they have no conflict of interest.

*Acknowledgements.*  The lidar measurements were performed during the *Polarstern* cruise PS96 funded by the Alfred-Wegner-Institute under *Polarstern* grant AWI_PS96_03. The research was funded by the SPP 1158 'Antarctic research' of the DFG (Deutsche Forschungsgemein-schaft) under grant HE 2740/19. The COSMO-CLM model was provided by the German Meteorological Service and the CLM commu-nity. Computing time was provided by DKRZ. AWS buoy sea ice measurements for 2016 are from http://www.meereisportal.de (funding:
REKLIM-2013-04). From all the software that was used we would like to highlight cdo, nco, R, RStudio and the R-packages data.table. We thank Niels Souverijns (KU Leuven, Belgium) and our colleagues in Trier for helpful discussions.



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



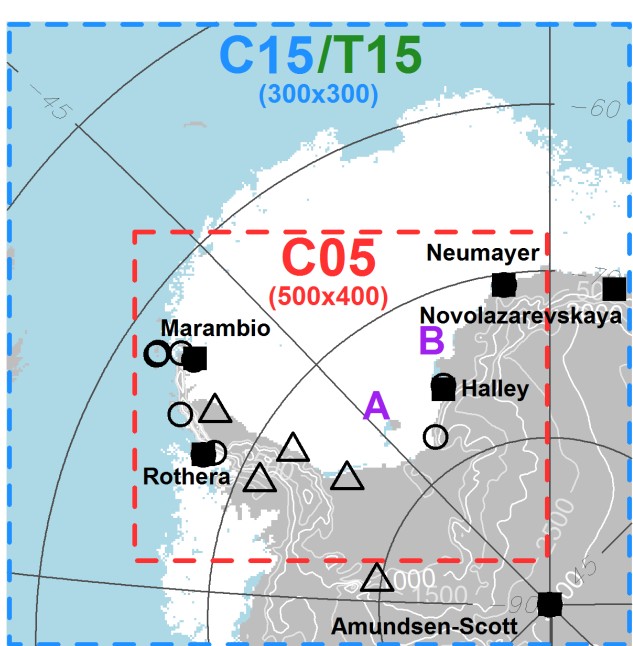

**Figure 1.** Overview of the C15/T15 (blue/green) and C05 (red) simulation domain, locations of 6 radio sounding stations (filled squares), surface stations (circles), automatic weather stations (triangles) and location of the RV *Polarstern* during our three case studies A and B (purple). Topography contours are plotted every 500 m and sea ice concentration >70% for the 1 June 2015 is shown in white. (Note that the T15 domain is the same as the C15 domain.)





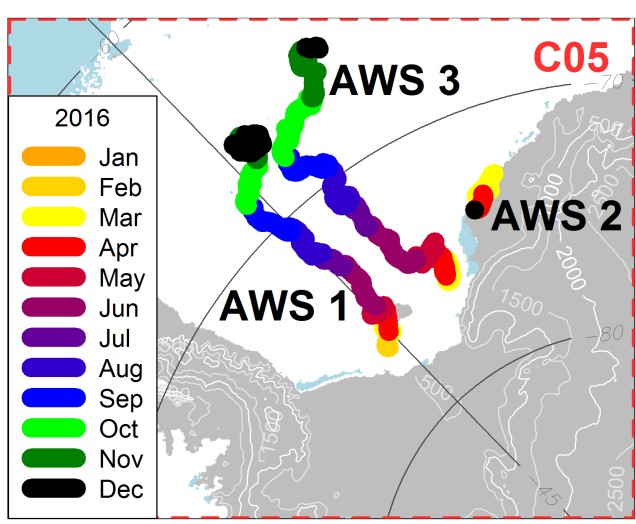

**Figure 2.** Overview of track of the three AWS buoy inside the C05 domain. Topography and sea ice concentration as in Fig. 1



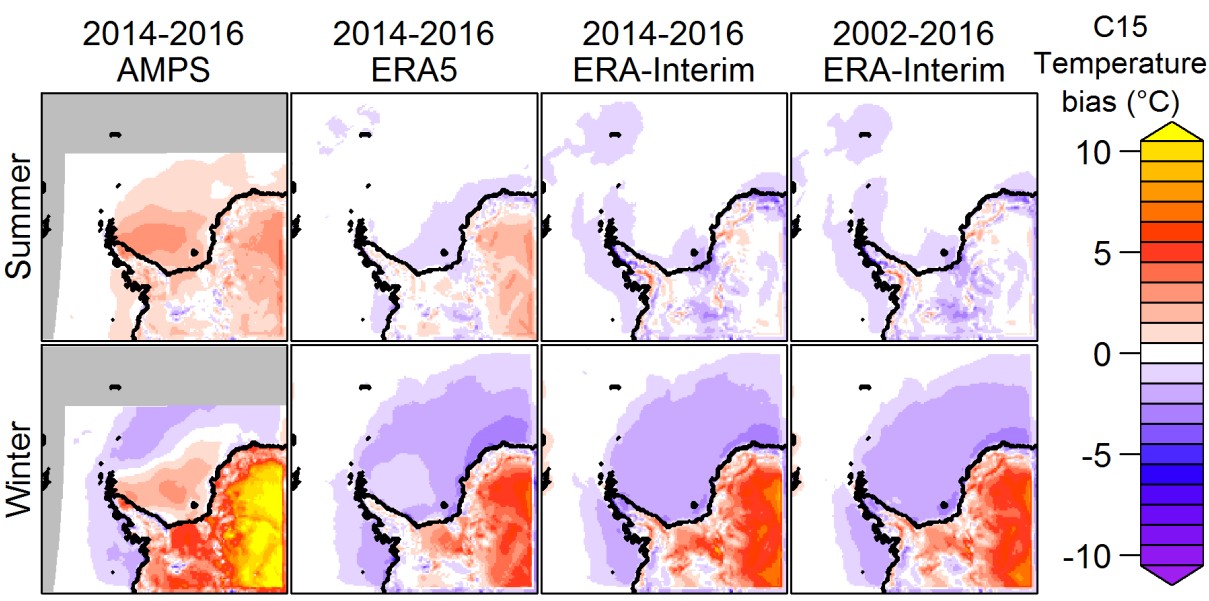

**Figure 3.** 2 m temperature bias of C15 compared to AMPS, ERA5 and ERA-Interim for the years 2014-2015 and to ERA-Interim for the year 2002-2016. Summer (ONDJFM, top) and winter (AMJJAS, bottom) are shown separately. The gray area is outside the AMPS-domain.





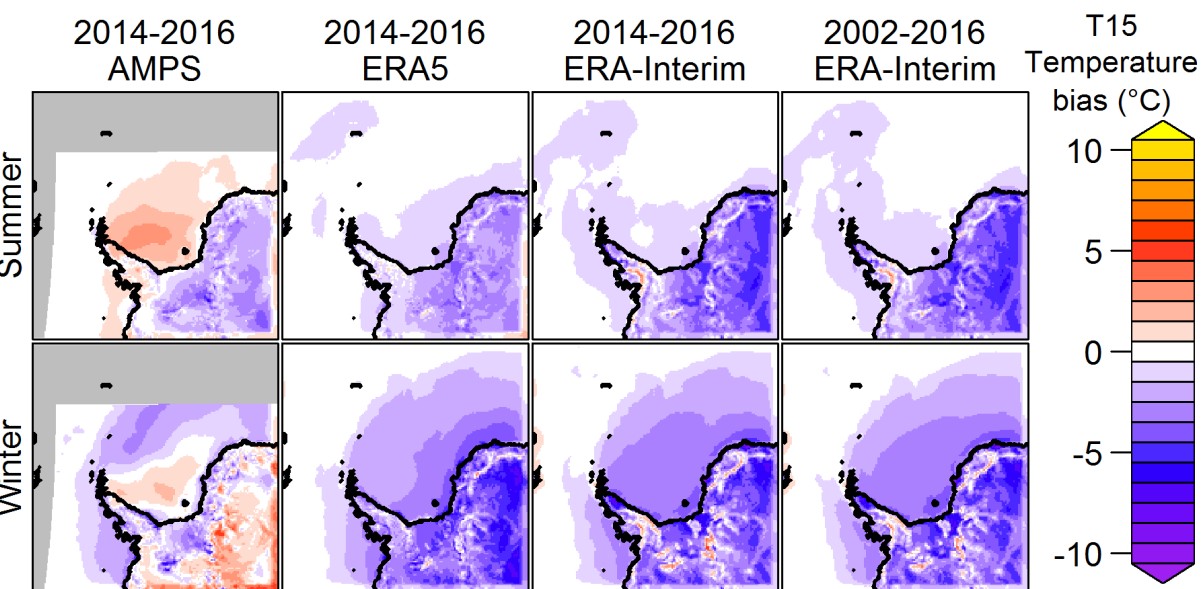

**Figure 4.** As Fig. 3 but for T15.



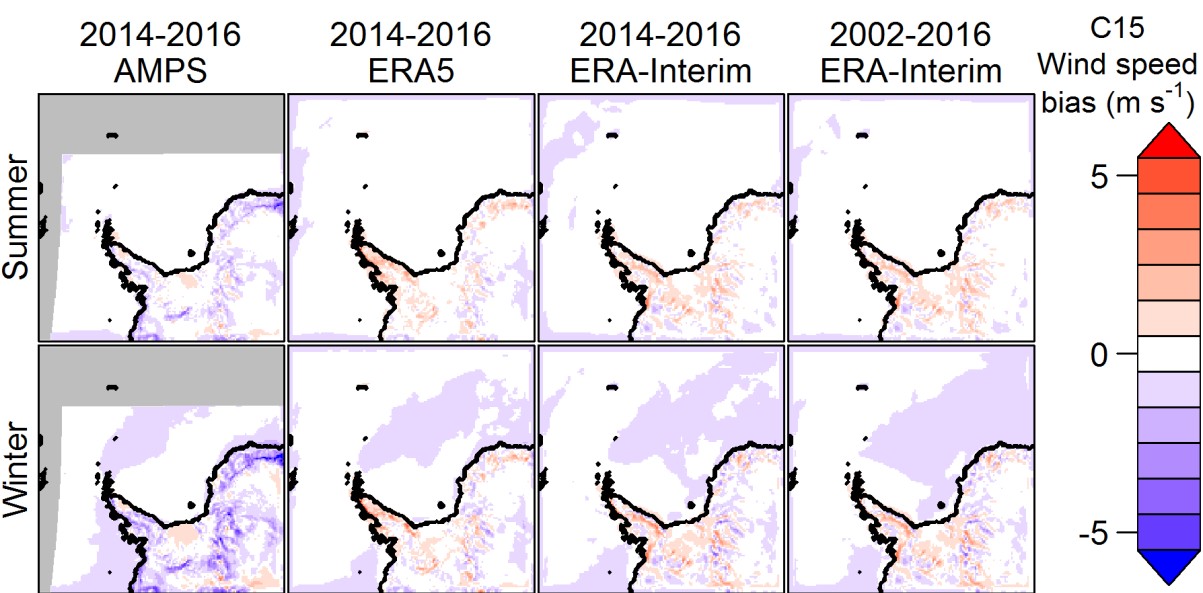

**Figure 5.** As Fig. 3 but for 10 m wind speed.



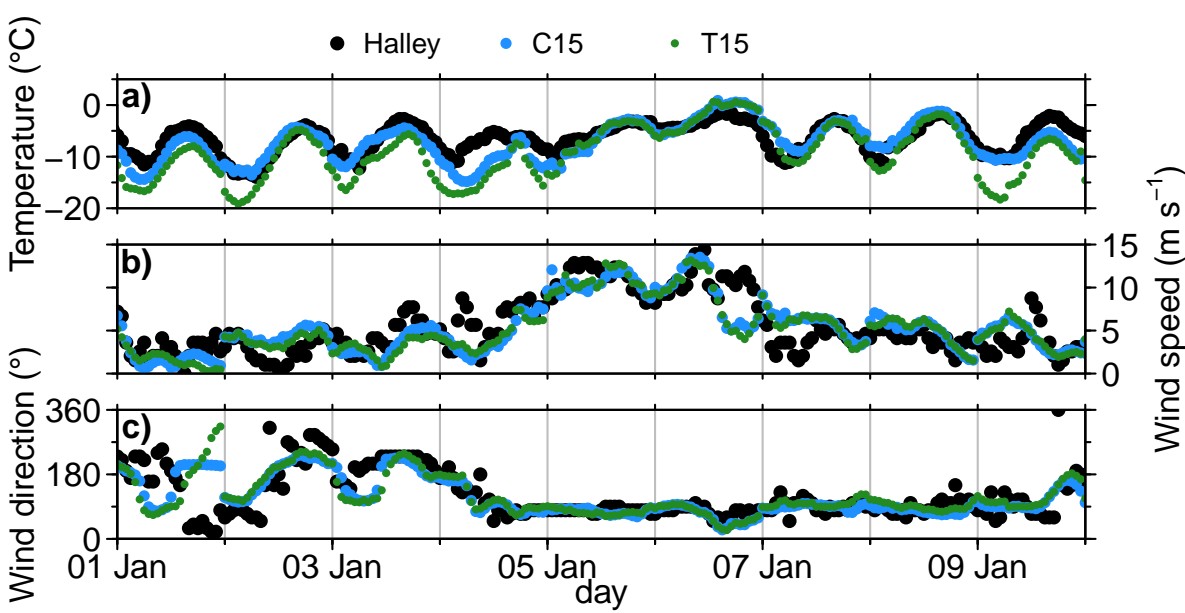

**Figure 6.** Comparison for Halley of 2 m temperature (a) 10 m wind speed (b) and 10 m wind direction (c) for station measurements (black), C15 (blue) and T15 (green) during January 2016. Vertical grey lines indicate the restart of the daily simulations.



**Figure 7.** CCLM 2 m temperature bias (a), RMSE (b) and correlation (c) for C05, C15 and T15 for different surface stations (see Table 4). Boxes indicate the 25/75% and whiskers the 10/90% quantile; the median is indicated by a black line inside the box. Statistics (bias, RMSE and correlation) are calculated for every month.

**Figure 8.** As Fig. 7 but for 10 m wind speed.



**Figure 9.** As Fig. 7 but for 10 m wind direction.

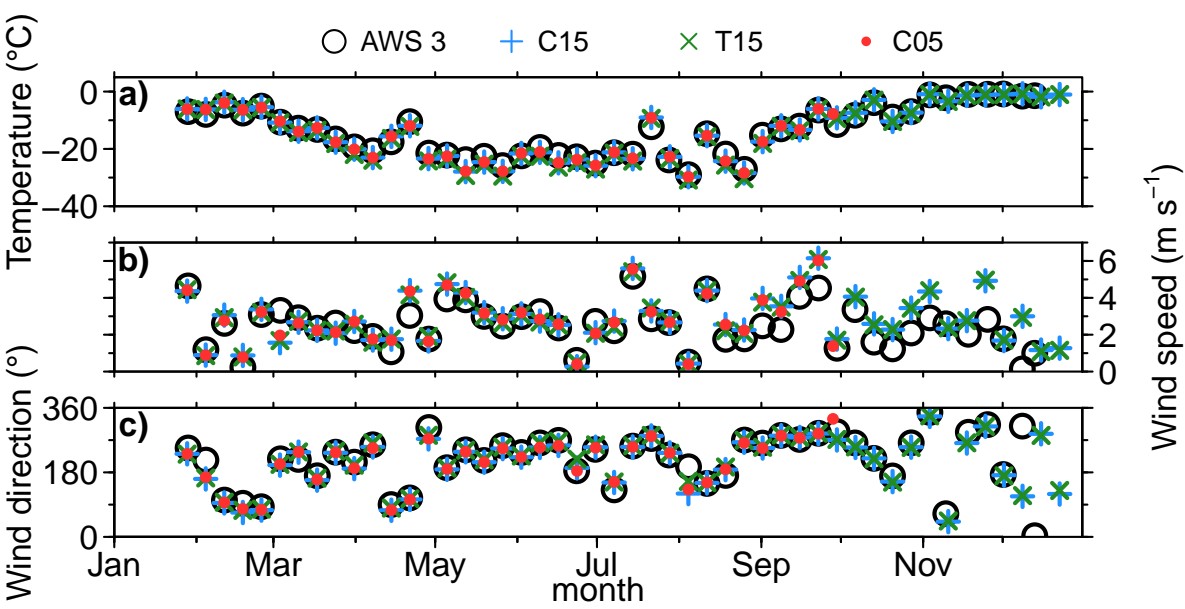

**Figure 10.** Weekly temperature (a), wind speed (b) and wind direction (c) in 2016 for AWS3 buoy in black (see Fig. 2), C15 (blue), T15 (green), C05 (red). The weekly mean was computed for zonal and meridional winds.



**Figure 11.** Temperature bias, RMSE (bottom axis) and correlation (top axis) of C15 during winter (solid line), C15 during summer (dashed line) and C05 during winter (dotted line) for the stations Marambio (a), Neumayer (b), Novolazarevskaya (c), Rothera (d), Halley (e) and Amundsen Scott (f).



**Figure 12.** Mean temperature of radio sounding (Raso, black), C15 (blue) and T15 (green) during winter (solid line) and during summer (dashed line) for the stations Marambio (a), Neumayer (b), Novolazarevskaya (c), Rothera (d), Halley (e) and Amundsen Scott (f). Note the different range on the x-axis for Amundsen Scott (f).



**Figure 13.** Like Fig. 11 but for wind speed.





**Figure 14.** Like Fig. 11 but for wind direction.



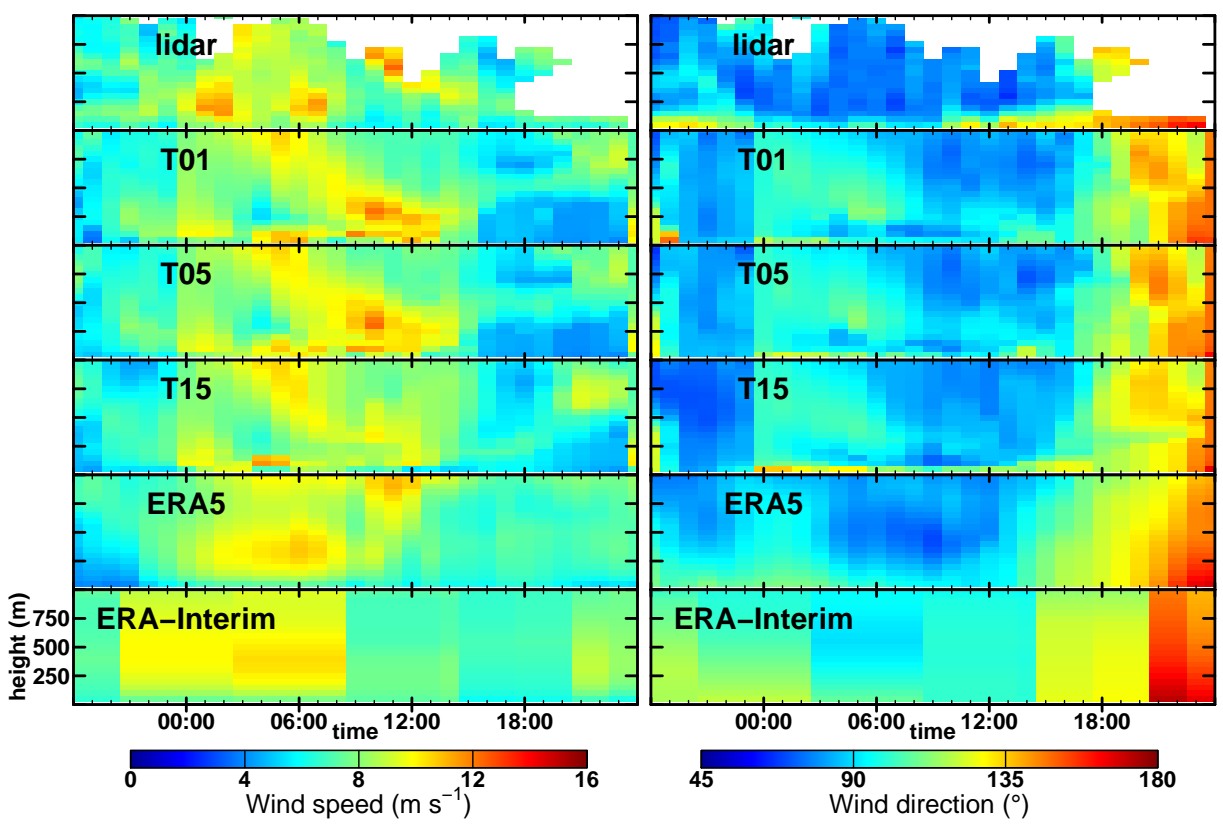

**Figure 15.** Time-height cross-sections for wind speed (left) and direction (right) for 16 Jan. 2016 18:00 UTC to 17 Jan. 2016 12:00 UTC.





**Figure 16.** Time-height cross-sections for wind speed (top) and direction (bottom) for 20 Jan. 00:00 UTC to 30 Jan. 00:00 UTC.





**Table 1.** Overview of the different simulations. The grid size for C01 and T01 was changed for each day with a minimal (maximal) size of 200 x 200 (353 x 464).

| | Simulations | | | | | |
|---|---|---|---|---|---|---|
| | **C15** | **T15** | **C05** | **T05** | **C01** | **T01** |
| Turbulence parameters changed | no | yes | no | yes | no | yes |
| Actual grid size (for rotated lat=0) | 13.88 km | | 5.55 km | | 1.11 km | |
| Grid size | 300 x 300 | | 400 x 400 | | > 200 x 200 | |
| Grid resolution (in rotated system) | 1.25° | | 0.05° | | 0.01° | |
| SSO used | yes | | yes | | no | |
| Period | Jan-Dec 2002-2016 | | Apr-Sept 2002-2016 | | Case Study only: Dec 2015,Jan 2016 | |





**Table 2.** Overview of surface parameters.

| Parameter | Value |
| --- | --- |
| Snow heat capacity (inland ice/ice shelf) | $0.73 \times 10^6$ J/m$^3$/K |
| Snow heat conductivity (inland ice/ice shelf) | 0.30 W/m/K |
| Albedo (glacial/shelf) | 0.80 |
| Heat capacity sea ice / snow on sea ice | $1.91/0.63 \times 10^6$ J/m$^3$/K |
| Heat conductivity sea ice / snow on sea ice | 2.26/0.75 W/m/K |
| Albedo (sea ice) | 0.75 |





**Table 3.** Overview of used sea ice concentration data in CCLM. Download for CERSAT/IFREMER under http://cersat.ifremer.fr/data/ and for Uni Bremen under https://seaice.uni-bremen.de/start/data-archive/.

| Satellite | Provider | Time period |
|-----------|----------|-------------|
| SSMI | CERSAT / IFREMER | 2002-01-01 till 2002-05-31 |
| AMSR | Uni Bremen | 2002-06-01 till 2011-10-04 |
| SSMI[a] | Uni Bremen | 2011-10-05 till 2012-07-23 |
| AMSR2 | Uni Bremen | 2012-07-24 till 2016-12-31 |

[a] There were two datasets of SSMI data based on different sensors (F17 and F18). A comparison for the overlapping periods of two month (August, September) with AMSR (2011) and AMSR2 (2012) were compared with the F17 and F18 SSMI data. Standard deviation was computed and it was found that F17 is closer to AMSR and F18 is closer to AMSR2, but overall F17 seemed to have less deviation in the area of interest. So only the F17 data was taken for Oct. 2011 – July 2012.



**Table 4.** Information of the surface stations. "yes" over land indicates that the surface type of the compared model grid point is land and not water. Years give the approximate data record length in years. AWS = automatic weather station, KGI = on King George Island

| | No. | Station | Longitude | Latitude | Height (m) | | | Land | | Years |
|---|---|---|---|---|---|---|---|---|---|---|
| | | | | | real | C05 | C15 | C05 | C15 | |
| inland | 1 | Amundsen Scott | 0.00 | -90.00 | 2835 | | 2796 | | yes | 15 |
| | 2 | Union (AWS) | -83.27 | -79.76 | 767 | | 1173 | | yes | 6 |
| east coast | 3 | Belgrano II | -34.62 | -77.87 | 256 | 235 | 388 | yes | yes | 14 |
| | 4 | Halley | -26.22 | -75.43 | 30 | 14 | 19 | yes | yes | 15 |
| | 5 | Neumayer | -8.25 | -70.67 | 50 | 35 | 36 | yes | yes | 14 |
| south pen. | 6 | Limbert (AWS) | -59.15 | -75.87 | 58 | 58 | 57 | yes | yes | 10 |
| | 7 | Butler (AWS) | -60.17 | -72.20 | 115 | 8 | 34 | yes | yes | 12 |
| | 8 | Fossil (AWS) | -68.28 | -71.32 | 66 | 182 | 279 | yes | yes | 10 |
| middle pen. | 9 | Rothera | -68.12 | -67.57 | 32 | 1 | 7 | yes | | 15 |
| | 10 | San Martin | -67.13 | -68.12 | 4 | 104 | 145 | | yes | 15 |
| | 11 | Vernadsky | -64.27 | -65.25 | 11 | 0 | 0 | | | 15 |
| | 12 | Larsen (AWS) | -61.47 | -67.00 | 43 | 37 | 31 | yes | yes | 10 |
| north pen. | 13 | Marambio | -56.72 | -64.23 | 198 | 0 | 3 | | | 15 |
| | 14 | Great Wall (KGI) | -58.97 | -62.22 | 10 | 37 | 61 | yes | | 14 |
| | 15 | Marsh(KGI) | -58.98 | -62.18 | 10 | 20 | 61 | yes | | 8 |
| | 16 | Bellingshausen (KGI) | -58.88 | -62.183 | 16 | 35 | 61 | | | 13 |
| | 17 | Esperanza | -56.98 | -63.40 | 13 | 212 | 201 | yes | yes | 15 |
| | 18 | Jubany/Carlini (KGI) | -58.63 | -62.23 | 4 | 72 | 119 | yes | yes | 15 |



**Table 5.** Information of radio sounding stations. Obs(servation) at UTC indicates the hour when the sounding was done. Interval shows usual time difference between the radio soundings (for 85% of all radio soundings). N indicates the number of radio soundings during 2002-2016.

|   | **Station** | **Longitude** | **Latitude** | **Height** (m) | | | **Obs at** | **Interval** | **N** |
|---|---|---|---|---|---|---|---|---|---|
|   |   |   |   | real | C05 | C15 | **UTC** | (in days) |   |
| a | Marambio | -56.63 | -64.23 | 198 | 16 | 5 | 12 | 1-5 | 1312 |
| b | Neumayer | -8.27 | -70.67 | 50 | 40 | 41 | 12 | 1 | 4765 |
| c | Novolazarevskaya | 11.83 | -70.77 | 119 |   | 216 | 0 | 1 | 5224 |
| d | Rothera | -68.13 | -67.57 | 16 | 6 | 12 | 12 | 1-3 | 954 |
| e | Halley | -26.66 | -75.58 | 30 | 34 | 35 | 12 | 1 | 4723 |
| f | Amundsen Scott | 0.00 | -90.00 | 2835 |   | 2800 | 0 | 1 | 5168 |



**Table 6.** CCLM temperature bias and RMSE for the three AWS buoy (see Fig. 2). N indicates the number of data points in 2016. Winter(month) are Apr.-Sept. and Summer(month) are Jan.-Mar. and Oct.-Dec.

| Name | N | Temperature bias | | | | Temperature RMSE | | | |
|------|------|--------|-----|--------|------|--------|-----|--------|------|
| | | Winter | | Summer | | Winter | | Summer | |
| | (hours) | C15 | T15 | C15 | T15 | C15 | T15 | C15 | T15 |
| AWS 1 | 7044 | -0.3 | -1.4 | 0.9 | 0.7 | 3.7 | 4.0 | 2.7 | 2.8 |
| AWS 2 | 7915 | 2.5 | 0.4 | 1.5 | 0.8 | 4.9 | 3.6 | 3.6 | 3.2 |
| AWS 3 | 6640 | -0.8 | -1.7 | 0.1 | -0.1 | 3.4 | 4.0 | 2.3 | 2.4 |





**Table 7.** As Table 6 but for wind speed.

| Name | N | Wind speed bias | | | | Wind speed RMSE | | | |
|---|---|---|---|---|---|---|---|---|---|
| | | Winter | | Summer | | Winter | | Summer | |
| | (hours) | C15 | T15 | C15 | T15 | C15 | T15 | C15 | T15 |
| AWS 1 | 7044 | 1.0 | 0.8 | 0.5 | 0.4 | 1.8 | 1.8 | 1.5 | 1.5 |
| AWS 2 | 7915 | 1.2 | 0.9 | 0.7 | 0.6 | 2.0 | 1.8 | 1.7 | 1.6 |
| AWS 3 | 6640 | 0.7 | 0.6 | 0.7 | 0.7 | 1.8 | 1.9 | 2.0 | 2.0 |





**Table 8.** Bias and RMSE for wind speed and direction compared to lidar measurements during December 2015 and January 2016.

| | | C01[a] | T01[a] | C05 | T05 | C15 | T15 | ERA5 | ERA-I |
|---|---|---|---|---|---|---|---|---|---|
| Wind speed (ms$^{-1}$) | Bias | 0.0 | 0.1 | 0.0 | 0.1 | -0.1 | 0.0 | -0.3 | -0.2 |
| | RMSE | 2.1 | 2.4 | 2.2 | 2.3 | 2.3 | 2.3 | 1.7 | 2.0 |
| Wind direction (°) | Bias | -3 | -5 | -3 | -5 | -5 | -5 | -1 | -2 |
| | RMSE | 28 | 30 | 32 | 29 | 32 | 30 | 22 | 29 |

[a] The runs with 1 km resolution were not performed for the whole period, but only for 37 days and 25 days (out of 39 days) for T01 and C01, respectively.