# Peer review of "Verification of the regional atmospheric model CCLM v5.0 with conventional data and lidar measurements in Antarctica"

_Geoscientific Model Development, 2019_

## Referee Comment (RC1) · Anonymous Referee #1 · 23 Oct 2019

This paper is a valid contribution to the scientific literature. It assesses the performance of the CCLM v5.0 for the Weddell Sea region. I have one general and a couple of specific concerns regarding the paper. I recommend the paper to be published only after these concerns have been adequately addressed.

Using Re-analyses data as reference in the validation is problematic. A recent paper by Gossart et al., (2019) for example shows strong warm biases in the interior of the continent in the different re-analyses products. It would be much better if only the observations were used for model validation. I recommend to remove the discussion

of the re-analyses and remove fig 3, 4 and 5 from the paper. If the authors feel strongly about keeping the re-analyses in their paper, it needs to be framed differently than is done now. From a comparison with observations – it can be investigated whether there is an added value in CCLM compared to the (driving) re-analyses. This can - for example – be done by extending figure 7, 8 and 9 and include the re-analyses here – if you think plots become too busy, you can differentiate winter and summer.

Related to that, I recommend to restructure the paper: 1) statistical analysis with station data, 2) comparison with Halley, 3) comparison with AWS3 buoys, 4) comparison with radiosondes 5) comparison with lidar

The methodology describing the sea ice is not completely clear: Is a fractional sea ice cover used in the model? This is particularly relevant when studying atmosphere-ice-ocean interactions – a goal that the authors have in mind. Can one grid box have sea ice classes of different thickness? Please clarify and also state the limitation associated with the assumptions made in the model.

The reduction of minimal diffusion coefficients for heat and momentum does indeed improve the performance in the interior, but deteriorates the performance on the ice shelves. Esp. in Fig 7 there is a strong increase in RMSE in winter over the east coast (and southern peninsula). This should be stated more clearly in the abstract and conclusions (in esp. the sentence 'Differences in other regions were small' is somewhat misleading). Do the authors have any idea how to improve the performance over the ice shelves? Is the albedo of the ice shelves correctly represented in the model and might deficiencies in albedo play a role?

Related to the previous point: Some information on the snow module should be included in the paper. Are albedo variations taken into account? How is the snow profile initialized and is this realistic? Even though this is a run in forecast mode, I assume that the surface is freely evolving. Is that right? Are snow temperatures drifting away from the forcing or is this not the case.

I am not sure if the forecast mode is the best when studying atmosphere ice ocean interactions – the sea ice cover in the driving re-analyses can be different than the observed cover and in that way processes related to atmosphere ice ocean interactions can be destroyed. A discussion on this topic in the conclusions / future work would be welcome. Moreover, it should be clearly indicated in abstract and conclusions that the model is used in forecast mode.

At the end of page 3 you describe you have a sea ice thickness of 0 m when the sea ice cover is 0-15%. I am not sure what this means – does it means that sea ice is simply ignored for these small fractional coverages? Although I did not dive into the reference, the value of 0.1 m for fractions between 15-70% seems very low to me. Can you somehow extent the argumentation on these values in the paper. Again this is quite relevant for the application that the authors have in mind.

Page 5 line 27 – you compare hourly averaged observations with grid box average instantaneous model output. You have to motivate this better – what is the typical advection speed and to which horizontal length scale does a time period of one hour correspond? Is it still possible to compare models and re-analyses with different resolutions if an evaluation is performed in this way. This point definitely needs more attention and a solid methodology needs to be presented and executed.

Figure 7, 8, and 9 are key figures to the paper, but difficult to interpret for the reader. Consider remaking them by plotting the box plots on a map, so that the reader directly knows to which station the comparison belong and is facilitated in the interpretation.

Consider switching Fig 11 with Fig 12.

Fig. 15 and 16: to facilitate the visual comparison, please remove the parts that are not measured with the lidar.

For the last part with the lidar comparison, also an evaluation of higher resolution integrations is added. Since sensitivity to resolution is small, I recommend to leave out this

comparison. It is sufficient to just make a note saying that decreasing the resolution to 5 or 1 km does not affect the wind patterns at the location of the lidar.

I suggest to merge the summary and conclusion and outlook section as there is some redundancy.

Reference:

Gossart, A., Helsen, S., Lenaerts, J.T M., Vanden Broucke, S., van Lipzig, N.P M., Souverijns, N. (2019). An Evaluation of Surface Climatology in State-of-the-Art Re-analyses over the Antarctic Ice Sheet. JOURNAL OF CLIMATE, 32 (20), 6899-6915. doi: 10.1175/JCLI-D-19-0030.1
* * *

---

## Referee Comment (RC2) · Anonymous Referee #2 · 3 Dec 2019

In this paper the authors present an interesting series of climate simulations for the Weddell Sea region of Antarctica run using the CCLM model version 5. They test two different turbulence parameterisations and use observations from manned and automatic weather stations as well as radiosondes and Lidar measurements to assess how well the model downscales Antarctic climate and the potential uses of CCLM in modelling atmosphere sea ice and ocean interactions.

This is an interesting paper presenting solid work evaluating a regional climate model in Antarctica with a range of data sources. It is well written and easy to follow and is

in fact a pretty good model of an evaluation paper for other groups who run climate simulations in the polar regions. I have a few comments that i think could help to improve the paper:

1) My main comment is the lack of detail in describing the model set-up. As one example, in section 2.1 (page 3 lines 6-11) the modifications to the turbulence parameterisation is discussed. The improvement in results shown in the figures is significant and it is therefore important, given also that this is a GMD paper, to be clear on exactly what was implemented.

2) Similarly when looking at the results compared with the station data it is not really clear what surface scheme is being used here as this may also have an impact on the biases shown.

3) The section on sea ice and SST setup is fairly clear but the authors do not mention if there is snow on sea ice and if/how this is dealt with in the model. Snow on sea ice can have important effects on the energy balance and it would be interesting to hear more abotu this aspect in CCLM.

4) It would similarly be useful to briefly discuss if/how similar this model version is with others that have been published recently such as by Gossart et al and Souverijns et al.

5) Is there nudging or relaxation in the domain or is forcing applied only on the boundaries? This has been shown by van Wessem to have a very significant affect on simulated Antarctic climate and details should be included if it is used

6) Figures 3- 5 showing the bias with respect to the different reanalyses is very interesting, in particular because it seems clear then reanalyses themselves disagree substantially in some locations. This point is not however well expressed within the paper and should be brought to the fore as it makes it challenging to verify against a reanalysis product if the reanalysis itself has some issues.

---

## Author Comment (AC1) · 16 Jan 2020

*Comment by referee #1*

*Response by authors*

*Changes in manuscript*

This paper is a valid contribution to the scientific literature. It assesses the performance of the CCLM v5.0 for the Weddell Sea region. I have one general and a couple of

specific concerns regarding the paper. I recommend the paper to be published only after these concerns have been adequately addressed.

Using Re-analyses data as reference in the validation is problematic. A recent paper by Gossart et al., (2019) for example shows strong warm biases in the interior of the continent in the different re-analyses products. It would be much better if only the observations were used for model validation. I recommend to remove the discussion of the re-analyses and remove fig 3, 4 and 5 from the paper. If the authors feel strongly about keeping the re-analyses in their paper, it needs to be framed differently than is done now. From a comparison with observations – it can be investigated whether there is an added value in CCLM compared to the (driving) re-analyses. This can – for example – be done by extending figure 7, 8 and 9 and include the re-analyses here – if you think plots become too busy, you can differentiate winter and summer.

We tried to take also comment 6 from referee #2 into account. We liked the idea of having Fig.3-5 at the beginning of the paper to give an impression of the performance of CCLM in comparison to AMPS and reanalyses. Because the later verifications are just single point observations that need to be seen in the right context. For example, the climatological difference of C15, T15, AMPS, ERA5/Interim are strongest over the east Antarctic plateau. But in this area we just have one station in the verifications shown later (surface and radio sounding).

We tried to address all raised concerns by moving the subsection 3.1 "Model and re-analyses" into another new section "Comparison" between the section "Data and Methods" and "Verification" and revising the section. It now states more clearly that AMPS and ERA5/Interim are not to be taken as verification data (we changed for example the phrasing "bias" to "difference" in the text and Fig. 3, 4 and 5).

We added "a short comparison to another model and reanalyses (section 3), then" to the introduction and renamed (and moved) the section to "Comparison with model and reanalyses".

[Figure]

In this section we changed "bias" to "difference" added two paragraphs: "Although a verification with measurements is preferable, due to the small number of stations in polar regions this is not possible for the whole model domain. A comparison to other simulations is therefore an addition to the evaluation, although it has its limits. Gossart et al. (2019) found that in some respects different reanalyses (including ERA5 and ERA-Interim) differ greatly for Antarctica and thus comparisons of CCLM with simulations should not be seen as a validation."

and "The study by Gossart et al. (2019) showed the largest differences in mean temperature between reanalyses over the interior Antarctica during winter (approx. 8 K) and that ERA and ERA-Interim are warmer than the observations. An evaluation of AMPS (Fig. A1 in Bromwich et al., 2005) showed only a small bias (down to -3 K) of AMPS in the interior Antarctica. Verifications using surface and radio sounding data (shown in section 4) confirm that C15 is too warm over the plateau and that this could be attributed to a too strong mixing in the surface boundary layer."

Related to that, I recommend to restructure the paper: 1) statistical analysis with station data, 2) comparison with Halley, 3) comparison with AWS3 buoys, 4) comparison with radiosondes 5) comparison with lidar

We actually wanted the comparison with Halley (Fig.6) as a case study before the statistical analysis. We moved the reanalysis/model section as proposed. (see comment above).

The methodology describing the sea ice is not completely clear: Is a fractional sea ice cover used in the model? This is particularly relevant when studying atmosphere-iceocean interactions – a goal that the authors have in mind. Can one grid box have sea ice classes of different thickness? Please clarify and also state the limitation associated with the assumptions made in the model.

We revised the section concerning the sea ice also with respect to the referee's later comment about ice thickness and fractions and referee #2 comment 3.

Concerning this comment we added to section 2.1:"A fractional sea ice cover is not used in the model, thus for each grid box there is only one value of sea ice thickness that is assumed to cover the whole grid box. Benefits of modelling a fractional sea ice cover are investigated in Gutjahr et al. (2016)."

The reduction of minimal diffusion coefficients for heat and momentum does indeed improve the performance in the interior, but deteriorates the performance on the ice shelves. Esp. in Fig 7 there is a strong increase in RMSE in winter over the east coast (and southern peninsula). This should be stated more clearly in the abstract and conclusions (in esp. the sentence 'Differences in other regions were small' is somewhat misleading). Do the authors have any idea how to improve the performance over the ice shelves? Is the albedo of the ice shelves correctly represented in the model and might deficiencies in albedo play a role?

The albedo of 0.8 is reasonable (see e.g. doi.org/10.1007/BF00120464), but most likely plays no role, as the RMSE is biggest in winter when no solar radiation is present. There is no general solution for improvement for the performance over ice shelves, since the Ronne-Filchner Ice Shelf (station 6) and Brunt Ice Shelf (station 4) show an increase in RMSE, but the Larsen Ice Shelf (station 12) shows a decrease for the new parameterization.

We removed the sentence "Differences in other regions were small." From the abstract and conclusion and added in the abstract ", but resulted in a negative bias for some coastal regions."

Related to the previous point: Some information on the snow module should be included in the paper. Are albedo variations taken into account? How is the snow profile initialized and is this realistic? Even though this is a run in forecast mode, I assume that the surface is freely evolving. Is that right? Are snow temperatures drifting away from the forcing or is this not the case.

We made revisions to include more details about this (we modified Table 2 accordingly).

[Figure]

We added in section 2.1: "The snow temperature profile is initialized with the forcing data, then the snow temperatures freely evolve. The surface albedo for inland ice and ice shelves is kept constant and has no seasonal variations. The albedo of sea ice is parameterized as a function of ice thickness and temperature by a modified Køltzow scheme (Køltzow, 2007) as described in Gutjahr et al. (2016a)."

We also added in section 2.1.: "For grid points with a sea ice thickness of 0.1 m the modified Køltzow scheme yields an albedo of 0.07 and we assume no snow cover. For a thickness of 1 m the albedo is 0.84 (for temperatures lower than -2°C) and fixed snow layer of 10 cm snow cover (Schröder et al. 2011) is assumed."

I am not sure if the forecast mode is the best when studying atmosphere ice ocean interactions – the sea ice cover in the driving re-analyses can be different than the observed cover and in that way processes related to atmosphere ice ocean interactions can be destroyed. A discussion on this topic in the conclusions / future work would be welcome. Moreover, it should be clearly indicated in abstract and conclusions that the model is used in forecast mode.

We use daily updated sea-ice concentrations from satellite data (6 km resolution) in the forecast mode, but we use a 6 hour spin up to allow for the atmosphere to adapt to the difference between the high-resolution sea ice data from satellite and the coarse-resolution temperatures from ERA-Interim.

We added "and used in forecast mode" in the abstract and "in forecast mode and" in the summary.

We added the sentence in section 2.1: "We used the first 6 hours as spin up to allow for the atmosphere to adapt to the difference between the high-resolution sea ice data from satellite and the coarse-resolution temperatures from ERA-Interim."

At the end of page 3 you describe you have a sea ice thickness of 0 m when the sea ice cover is 0-15%. I am not sure what this means – does it means that sea ice is

simply ignored for these small fractional coverages? Although I did not dive into the reference, the value of 0.1 m for fractions between 15-70% seems very low to me. Can you somehow extent the argumentation on these values in the paper. Again this is quite relevant for the application that the authors have in mind.

Yes, with an ice thickness 0 m we meant open water. We corrected it. The value of 0.1 m for fractions between 15-70% is justified by studies of sea ice thickness in polynyas (see e.g. https://doi.org/10.5194/tc-10-2999-2016). A threshold of 70% is a well-accepted value for the detection of polynyas. Many observational studies have shown that only a small area of wintertime polynyas is ice-free. We assume this for 0-15% sea ice concentration, as 15% is also a common threshold for the ice edge.

We changed the sentence to: "Grid points with a sea ice concentration of 0-15% are set to open water. For 15-70% a sea ice thickness of 0.1 m is assumed (see e.g. Gutjahr et al.,2016b). For 70-100% we assume a thickness of 1 m, which is a reasonable estimate for the Weddell Sea (see Kurtz and Markus, 2012)."

Page 5 line 27 – you compare hourly averaged observations with grid box average instantaneous model output. You have to motivate this better – what is the typical advection speed and to which horizontal length scale does a time period of one hour correspond? Is it still possible to compare models and re-analyses with different resolutions if an evaluation is performed in this way. This point definitely needs more attention and a solid methodology needs to be presented and executed.

This is a general question of comparison of model data with observations at a point.

As mentioned, model data represent volume averages over a grid box. If you only look at the advection speed, then a 10 m/s speed would correspond to a distance of 36 km, which is about two times the horizontal grid spacing for C15/T15 and seven times for C05/T05. On the other hand, the horizontal grid distance is not the resolution of the model in the sense of the representation of processes. Using spectral analysis methods for CCLM it was found that the effective model resolution is at least 5-7 times the

horizontal grid spacing (Zentek et al. 2016, 10.1175/JCLI-D-15-0540.1). An instanta-neous model output at a grid point is therefore always a smoothed value over a much larger scale than the grid distance. For the lidar data, it is the other way round. Here the sampling for a single measurement is a few seconds (and thus contains also tur-bulence), and a wind profile corresponds to a time scale of 1-2 minutes. Therefore, we averaged the profiles over time in order to remove some of the small-scale variability. The same problems occur when model data is compared e.g. to radio soundings. It is generally assumed that the time of the ascent is e.g. 1200 UTC, but in reality the ascent is over about two hours and the 1200 UTC sonde is launched much earlier. It is also not clear, if the synoptic observations and AWS data used for the comparison are really averages over an hour or if they are e.g. 10min averages every hour.

In summary, we follow the methodology of previous verification studies. We rephrased the sentence, on why we averaged the lidar data, but we will not discuss all other possible problems.

Before: "Further note that the lidar data is an average over one hour around every full hour, which smooths the data and makes it better comparable to the simulation data that represent the wind average over the whole model grid box." Now:

"Further note that the lidar data is an average over one hour around every full hour, which removes small-scale variability as the single measurements were done approx-imately every 15 min for 1-2 min. This makes it better comparable to the simulation data, because although the output is instantaneous, it unlikely shows turbulence on such a small scale as it always represent the wind average over the whole model grid box."

Figure 7, 8, and 9 are key figures to the paper, but difficult to interpret for the reader. Consider remaking them by plotting the box plots on a map, so that the reader directly knows to which station the comparison belong and is facilitated in the interpretation.

We adapted the map (Fig. 1) by replacing the symbols with the stations numbers used

in Fig. 7, 8 and 9.

Consider switching Fig 11 with Fig 12.

We switched them.

Fig. 15 and 16: to facilitate the visual comparison, please remove the parts that are not measured with the lidar.

By removing these parts, some information would also be lost (e.g. the wind maxima around 11:00 UTC in 750 m height in Fig.15 that is present in ERA5 but not in ERA-Interim). We compromised by drawing a contour, thus enhancing the visual comparison.

For the last part with the lidar comparison, also an evaluation of higher resolution integrations is added. Since sensitivity to resolution is small, I recommend to leave out this comparison. It is sufficient to just make a note saying that decreasing the resolution to 5 or 1 km does not affect the wind patterns at the location of the lidar.

Generally we agree. On the other hand, not much space is gained by leaving that out these results and readers prefer to see the results directly, so we kept it.

I suggest to merge the summary and conclusion and outlook section as there is some redundancy.

The summary is more detailed while the conclusion and outlook section is more general (and very short). Thus we think it is justified to stay with the two separate sections.

Reference: Gossart, A., Helsen, S., Lenaerts, J.T M., Vanden Broucke, S., van Lipzig, N.P M.,Souverijns, N. (2019). An Evaluation of Surface Climatology in State-of-the-Art Reanalyses over the Antarctic Ice Sheet. JOURNAL OF CLIMATE, 32 (20), 6899-6915. doi: 10.1175/JCLI-D-19-0030.1

[Figure]

**C15/T15**
**(300x300)**

**C05**
**(500x400)**

Neumayer

Novolazarevskaya
**B**

18 17
16 Marambio
15 13
14

11 12
10 7
9
Rothera 8 6

**A**

4 Halley

2 1000
1500

Amundsen-Scott

**Fig. 1.**

Wind speed (m s$^{-1}$)

Wind direction (°)

**Fig. 2.**

---

## Author Comment (AC2) · 16 Jan 2020

*Comment by referee #2*

*Response by authors*

*Changes in manuscript*

In this paper the authors present an interesting series of climate simulations for the Weddell Sea region of Antarctica run using the CCLM model version 5. They test

two different turbulence parameterisations and use observations from manned and automatic weather stations as well as radiosondes and Lidar measurements to assess how well the model downscales Antarctic climate and the potential uses of CCLM in modelling atmosphere sea ice and ocean interactions.

This is an interesting paper presenting solid work evaluating a regional climate model in Antarctica with a range of data sources. It is well written and easy to follow and is in fact a pretty good model of an evaluation paper for other groups who run climate simulations in the polar regions. I have a few comments that i think could help to improve the paper:

1) My main comment is the lack of detail in describing the model set-up. As one example, in section 2.1 (page 3 lines 6-11) the modifications to the turbulence parameterisation is discussed. The improvement in results shown in the figures is significant and it is therefore important, given also that this is a GMD paper, to be clear on exactly what was implemented.

We changed the paragraph:

Before: "In the T15 simulation, the minimal diffusion coefficients for heat and momentum were lowered (from 0.4 to 0.01 m2s−1) to allow for a very stable boundary layer over the Antarctic ice sheet during winter. Further, the parametrization of the impact of the inhomogeneity of the surface on the turbulent kinetic energy (TKE) was modified. These modifications are based on the studies of Cerenzia et al. (2014), Hebbinghaus and Heinemann (2006) and Souverijns et al. (2019)."

Now: "These modifications are based on the studies of Cerenzia et al. (2014), Hebbinghaus and Heinemann (2006) and Souverijns et al. (2019). In the standard version of CCLM, the diffusion coefficients for heat and momentum are restricted to the minimal value of 0.4 m2s−1. In the T15 simulation, these minimal diffusion coefficients were set to 0.01 m2s−1 to allow for a very stable boundary layer (SBL) over the Antarctic ice sheet during winter. Further, the standard setup of CCLM uses a parameterization

of the impact of the inhomogeneity of the surface via the energy transfer from subgrid scale eddies on the turbulent kinetic energy (TKE). Since this leads to an overestimation of the TKE in the SBL (Cerenzia et al. 2014), this parameterization was removed in the T15 runs."

2) Similarly when looking at the results compared with the station data it is not really clear what surface scheme is being used here as this may also have an impact on the biases shown.

We added to section 2.1: "Over land, we use the standard land surface model of CCLM (TERRA, see archived documentation under zenodo (Zentek, 2019). The soil model has eight layers (down to 15 m) and allows for an additional snow layer on top of the soil, which varies with precipitation and sublimation. For the land ice regions, soil was replaced by snow using the parameters listed in Table 2."

3) The section on sea ice and SST setup is fairly clear but the authors do not mention if there is snow on sea ice and if/how this is dealt with in the model. Snow on sea ice can have important effects on the energy balance and it would be interesting to hear more abotu this aspect in CCLM.

We revised the section concerning the sea ice also with respect to the comments of referee #1.

Concerning this comment we added in section 2.1: "The snow temperature profile is initialized with the forcing data, then the snow temperatures freely evolve. The surface albedo for inland ice and ice shelves is kept constant and has no seasonal variations. The albedo of sea ice is parameterized as a function of ice thickness and temperature by a modified Køltzow scheme (Køltzow, 2007) as described in Gutjahr et al. (2016a)."

We also added in section 2.1.: "For grid points with a sea ice thickness of 0.1 m the modified Køltzow scheme yields an albedo of 0.07 and we assume no snow cover. For a thickness of 1 m the albedo is 0.84 (for temperatures lower than -2°C) and fixed snow

layer of 10 cm snow cover (Schröder et al. 2011) is assumed."

4) It would similarly be useful to briefly discuss if/how similar this model version is with others that have been published recently such as by Gossart et al and Souverijns et al.

We added to section 2.1: "Lastly we want to point out some differences between the present model setup and the setup of Souverijns et al. (2019), as they also used the CCLM model for simulations in the Antarctic. Souverijns et al. (2019) use CCLM with the community land model CLM (van Kampenhout et al., 2017), while we use default land surface model of CCLM with the adaptions described above. While we use daily high-resolution (6 km) sea ice data from satellites, they use coarse resolution ERA-Interim data (80 km) for the sea ice. In addition, they use only the standard one-layer sea ice model of CCLM."

5) Is there nudging or relaxation in the domain or is forcing applied only on the boundaries? This has been shown by van Wessem to have a very significant affect on simulated Antarctic climate and details should be included if it is used

We appended another sentence to the last changes concerning the last comment: "They also ran CCLM in climate mode and applied spectral nudging, while we used forecast mode with a restart every day and applied forcing only at the boundaries."

6) Figures 3- 5 showing the bias with respect to the different reanalyses is very interesting, in particular because it seems clear then reanalyses themselves disagree substantially in some locations. This point is not however well expressed within the paper and should be brought to the fore as it makes it challenging to verify against a reanalysis product if the reanalysis itself has some issues.

We tried to take also the first comment from referee #1 into account and made an effort to express more clearly the issues of reanalyses and discuss the comparison in a way that gives it more importance on the one hand (e.g. by moving it to a new separate section of its own), but also raises awareness of the limits of such a comparison on the

other hand.

We added "a short comparison to another model and reanalyses (section 3), then" to the introduction and renamed (and moved) the section to "Comparison with model and reanalyses".

In this section we changed "bias" to "difference" added two paragraphs: "Although a verification with measurements is preferable, due to the small number of stations in polar regions this is not possible for the whole model domain. A comparison to other simulations is therefore an addition to the evaluation, although it has its limits. Gossart et al. (2019) found that in some respects different reanalyses (including ERA5 and ERA-Interim) differ greatly for Antarctica and thus comparisons of CCLM with simulations should not be seen as a validation."

and "The study by Gossart et al. (2019) showed the largest differences in mean temperature between reanalyses over the interior Antarctica during winter (approx. 8 K) and that ERA and ERA-Interim are warmer than the observations. An evaluation of AMPS (Fig. A1 in Bromwich et al., 2005) showed only a small bias (down to -3 K) of AMPS in the interior Antarctica. Verifications using surface and radio sounding data (shown in section 4) confirm that C15 is too warm over the plateau and that this could be attributed to a too strong mixing in the surface boundary layer."

---

## Author Response (AR2)

**Topical Editor Decision: Publish subject to technical corrections**
(04 Mar 2020) by Fabien Maussion
Comments to the Author:
Dear Authors,

thank you for your revised manuscript. Based on the two reviews and my own assessment of the manuscript, I am happy to accept this paper for final publication in GMD, after a couple of technical comments (see below) are addressed. I spotted some typos which should be corrected during proof reading.

My apologies for the unusually long time needed for this review process.

Best regards,

Fabien Maussion

P1 L3: "and used in forecast mode" - I'm not sure that "forecast mode" is a generally accepted term for this kind of simulation (I personnally know and use "daily renitialisation" for example, and other groups might use yet another term). Maybe add "used in forecast mode (suite of consecutive 30H long simulations with 6H spin-up)."
We added it to the abstract.

P1 L15: "These results encourage for further studies using CCLM data for the regional climate in the Antarctic at high resolutions and the study of atmosphere-ice-ocean interactions processes" -> sentence difficult to read, you may consider rephrasing it.
We changed it to:
"Based on these encouraging results CCLM at high resolution will be used for the investigation of the regional climate in the Antarctic and atmosphere-ice-ocean interactions processes in a forthcoming study."

P3 L23: "temperatures freely evolve during the 30H simulation". I think that this precision is important, since the reset of soil properties to coarse and possibly biased input data is a possible drawback of the daily reinitialisation strategy.
As the model-restart is described in the paragraph just above we feel no changes are necessary.

P6 L25: Title "Comparison with AMPS and ERA" for consistency with title 2.2
Changed

P6 L29: "differ greatly between each other in Antarctica and thus comparisons of CCLM with these data should not be seen as a validation."
Changed

P7 L17: "ParameterizationS"
Changed

P7 L25: title: consider changing to "Validation" or (in my opinion better) "Comparison to observations"
Changed title to "Comparison to observations"
also changed title 2.5 and 4.3 to "Wind lidar"
and title 2.3 to "AWS and surface stations"
to increase consistency

P7 L31: "An" -> A
Changed

P8 L22: "In the last section (3)" -> "In Sect. 3"
Changed to "In section 3"

Code & Data availability section: please check that no changes in code or data were necessary after the revisions. I assume that at least the plotting scripts need an update?
Yes, changes are limited to the plotting scripts. We upload the new version.

[revised manuscript text omitted]